# Manufacturing PLA/PCL Blends by Ultrasonic Molding Technology

**DOI:** 10.3390/polym13152412

**Published:** 2021-07-22

**Authors:** Inés Ferrer, Ariadna Manresa, José Alberto Méndez, Marc Delgado-Aguilar, Maria Luisa Garcia-Romeu

**Affiliations:** 1Departament d’Enginyeria Mecànica i de la Construcció Industrial, EPS—Universitat de Girona, c/M. Aurèlia Capmany, 61-17003 Girona, Spain; manresa.ariadna@gmail.com (A.M.); mluisa.gromeu@udg.edu (M.L.G.-R.); 2LEPAMAP-PRODIS Research Group, University of Girona, c/M. Aurèlia Capmany, 61-17003 Girona, Spain; jalberto.mendez@udg.edu (J.A.M.); m.delgado@udg.edu (M.D.-A.)

**Keywords:** ultrasonic molding, ultrasonic plasticizing, polymeric blends, biodegradable polymers, bioabsorbable polymers

## Abstract

Ultrasonic molding (USM) is a good candidate for studying the plasticization of polymer mixtures or other composite materials due to either the little amount of material needed for processing, low waste or the needed low pressure and residence time of the mold. Thus, the novelty of this research is the capability of USM technology to process PLA/PCL blends and their corresponding neat materials, encompassing all the production stages, from raw material to the final specimen. The major findings of the work revealed that the thermal properties of the blends were not affected by the USM process, although the crystallinity degree experienced variations, decreasing for PLA and increasing for PCL, which was attributed to the crystallization rate of each polymer, the high process speed, the short cooling time and the small particle size. The employed ultrasonic energy increased the molecular weight with low variations through the specimen. However, the degradation results aligned with the expected trend of these material blends. Moreover, this study also showed the effect pellet shape and dimensions have over the process parameters, as well as the effect of the blend composition. It can be concluded that USM is a technology suitable to successfully process PLA/PCL blends with the correct determination of process parameter windows.

## 1. Introduction

In recent decades, research of novel materials for advanced fields, such as medical applications, has notably increased to replace or recover the tissue functions of the human body [1] or improve quality of life [2]. Some relevant biomaterial applications include tissue engineering and regenerative medicine [3], surgical implants or bone-fixing devices in orthopedic applications [4], porous structures in tissue engineering [5], implantable matrices for controlled drug release inside the body [6] or absorbable sutures [7], among others.

The use of synthetic polymers in the medical industry has been extended due to their capability to fulfill many functional objectives required in medical devices, namely biocompatibility, biodegradability and mechanical properties [1], in addition to bioactivity or antimicrobial properties when particle-reinforced polymers or coatings are added [7]. According to Middleton and Tipton [6], the ideal polymer must be (i) metabolized by the body after its purpose without leaving a trace (biodegradable and bioabsorbable), (ii) easily sterilized, (iii) mechanically resistant, (iv) easy to process to its final form and, in addition, (v) it should not cause inflammatory or toxic effects disproportionate to its beneficial effect.

Biodegradability is a huge advantage for medical developments, as it prevents the removal of implants or other fixation devices after their shelf life. However, biodegradable materials usually exhibit limited mechanical performance and need to be combined or blended with more resistant polymers in order to fulfill the expected functionality. Thus, polymer blends are interesting due to the easy, affordable and scalable way to enhance the properties of pristine homopolymers. Indeed, aliphatic polyesters are very attractive for medical applications due to their high level of biocompatibility, acceptable degradation rates and great versatility in physical and chemical properties [8]. In particular, polyesters derived from glycolic acid (PGA), polylactic acid (PLA), polycaprolactone (PCL) and p-dioxanone (PPDX) are used as surgical sutures or implants with temporary, mechanically therapeutic functions [7] (The list of abbreviations can be found in Abbreviations).

In recent years, PLA has been reported to be one of the most promising bio-based and biodegradable polymers due to the wide range of applications in which it has potential use, especially packaging or as a potential substitute for polyolefin [9,10]. In addition, PLA has been deeply studied for its potential use in medical applications such as tissue engineering and bone-fixation devices, mainly due to its biocompatibility, biodegradability and high stiffness [11]. More recently, the potential of PLA to be used as feedstock material for fused deposition modeling equipment, commonly referred to as 3D printers, has attracted the interest of different industries [12,13]. The high glass transition temperature (T_g_) of PLA results in excessive brittleness, limiting its use in practical applications. In this sense, toughening PLA is one of the main challenges described in the literature, performed by means of several methods, including polymer modification and blending with other polymers or fillers [14,15].

The mixing of two or more polymers is a well-established method to obtain new structures and materials with desired characteristics as a result of the combination of properties [11,16]. Several studies have focused on the development of PLA-based blends with other biopolymers, such as thermoplastic starch (TPS) [17], polyhydroxybutyrate (PHB) [18], poly(butylene adipate-*co*-terephthalate) (PBAT) [10], polycaprolactone (PCL) [19,20] or even polyamide 11 (PA11) [21]. Out of these matrices, PCL has been deeply studied to tailor the properties of PLA, mainly due to its biodegradability and the interesting synergies that can be obtained. PCL is nontoxic, easily obtainable, biocompatible and biodegradable, and it is produced from nonrenewable resources. In addition, its rubbery amorphous phase at room temperature, low melting temperature (around 60 °C) and high toughness make PCL an interesting polymer to be combined with PLA [15]. PCL has potential applications in long-term implants and controlled drug release applications, mainly due to its biocompatibility and processability [22,23].

The main reason behind blending PLA with PCL falls on the development of improved toughness materials while retaining their biocompatibility and environmentally friendly characteristics [11]. While it is true that the low melting temperature of PCL usually limits the application of PLA/PCL blends in high-temperature applications, the softening temperature at high PLA contents may be governed by the T_g_ of PLA. Blending PLA with PCL usually decreases the brittleness of PLA, providing a wider range of potential applications due to the improvement in toughness, but it has been extensively reported that the incorporation of PCL may decrease the tensile and flexural strength of the materials [24,25]. PLA and PCL have been reported to have low phase miscibility, and several coupling agents have been incorporated to improve the stress transfer from one matrix to the other [19,26]. However, the incorporation of such compatibilizers has not been found to provide substantial enhancement.

In terms of manufacturing, injection molding has been the predominant manufacturing process for processing PLA and PCL blends; however, focusing on medical applications, they often require small production batches with miniature/microscale components at an affordable cost. In that sense, ultrasonic molding (USM) technology has significant advantages, allowing the fabrication of polymeric microcomponents with: (a) good replication, repeatability and high precision of microcharacteristics [27,28]; (b) a low ratio of material waste and energy saving, as only the material required per a single shot is melted, leading to the efficient use of raw materials and energy, with a positive impact on cost and environmental impact [29,30]; (c) short temperature exposure time of polymers, lasting only a few seconds [31,32]; and, finally, (d) small batches of production at an affordable cost, especially when high-performance applications are required [29,33].

USM technology applied to polymers is currently under investigation with different approaches. The first approach focuses on analyzing the processability of homopolymers, such as polypropylene (PP) [30], polylactic acid (PLA) [34,35], polyamides (PAs) [36], polystyrene (PS) [27], polyphenylsulfone (PPSU) [37], polyether ether ketone (PEEK) [38] and cyclic olefin copolymer (COC) [39], among others. In the second approach, the technology is used to process components of polymeric blends, composites or nanocomposites. Planellas et al. [40] focused on studying the dispersion of nanoclays in polylactide (PLA) and polybutylene succinate (PBS) matrices using ultrasound technology. Olmo et al. [41] proved that ultrasound micromolding is a suitable technology to process poly(ε-caprolactone), as well as its nanocomposite with multiwalled carbon nanotubes (MWCNTs) (loads of 5 wt %), and Sánchez-Sánchez et al. [42] confirmed its feasibility for manufacturing ultrahigh molecular weight polyethylene (UHMWPE)/graphite composites (1 wt %, 5 wt % and 7 wt %). The third approach is based on understanding the plasticization mechanisms and the ultrasonic behavior of the melt polymer. In that sense, Jiang et al. [43] conducted research comparing the plasticization process caused by heat transfer or ultrasonic energy. Later on, they numerically and experimentally studied the characteristics of viscoelastic heating [44] and interfacial friction heating mechanisms [45] during the ultrasonic plasticization process. Lastly, they analyzed the influence of the ultrasonic process on polymer fluidity [46]. These three approaches are relevant to our work due to their integration of process parameters, the plasticizing effect and blend processing.

USM can process an extended range of polymers, and the literature reveals a great potential for manufacturing expensive and sensitive polymers used in biomedical applications, such as drug delivery or tailored implants [29], where polymeric blends and composites are important objectives to fulfill with essential medical requirements such as biodegradability or bioactivity. However, there is a gap in PCL processing, its combination with PLA and the simulation of the volumetric degradability under similar conditions to the human body. Considering the above, the aim of the present work is to analyze the feasibility of manufacturing PLA/PCL blends by USM technology; find the process parameter windows to obtain complete specimens or parts; analyze the effect of pellet shapes and their dimensions on the ultrasonic energy required during the process, as well as their influence on the portion blends; and, finally, characterize the specimens by means of thermal and chemical characteristics and their capacity to degrade under acid and basic conditions for two, four and six weeks.

## 2. Ultrasonic Molding Technology

USM is based on the use of high-power ultrasonic energy for melting a small amount of material to shape parts with reduced size. Although several machine configurations [33] exist, the machine used in this work (Sonorus 1G) (Figure 1a) uses a sonotrode to melt the material and a plunger to push it into the mold cavity to apply the specific force required for packing it into the mold (Figure 1b). As explained in detail in the literature [29,33], the process consists of six main stages: feeding, compaction, preheating, melting, injection/packing and cooling.

Table 1 shows the set of process parameters that can be controlled at each stage, labeled by the marker ✓. The black square means that the parameter is not applicable. Here, the strokes number (St) is the number of impacts that the plunger applies to the pellets by moving up and down, controlled by specific force (F) and speed (v). When compacting the pellets with each other, the surface in contact with the sonotrode is flattened, avoiding tangential forces to the sonotrode that can displace it laterally and affect its vibration. Moreover, the trapped air is eliminated, keeping the heat transfer more homogeneous and, therefore, the fusion process more stable. The vibrational amplitude (A) and its application time (t) become the ultrasonic energy applied to the material both for preheating and melting. In the melting process, the sonotrode melts the material as the plunger pushes it into the mold cavity. Here, the adjustment of process parameters is a critical step to guarantee that no solid material is dragged into the mold cavity, particularly when processing small pellets. The plunger force is the fixed push that the pellets receive, the speed is related to its velocity and the plunger displacement (D) is the distance that the plunger travels to move the melted material and fill the mold cavity. Finally, applying the required holding force to pack it during the required time, the shrinkage is compensated, and the geometrical dimensions of the manufactured parts can be guaranteed. (The list of abbreviations can be found in Abbreviations).

## 3. Material, Geometry and Methods

### 3.1. Material and Geometry

The experimental material used was Ingeo™ Biopolymer 3251D poly(lactic acid), supplied by Natureworks LLC, and Capa^®^ 6500 Polycaprolactone poly(ε-caprolactone), provided by Ingevity^TM^, which were blended as described below, forming three polymer blends (%PLA/%PCL): 90PLA/10PCL, 80PLA/20PCL and 70PLA/30PCL. The geometry was a standard tensile specimen, according to UNE-EN ISO 20753:2008, with designation A15 but with 1 ± 0.1 mm of thickness (Figure 1c).

### 3.2. Experimental Approach

The experimental approach is divided into four main steps, which are summarized in the manufacturing process chain pictured in Figure 2. In the first step, specimens of the PLA and PCL commercial pellets were manufactured to establish process parameter windows that could be used to manufacture the specimen blends, but some preliminary trials were performed to set the starting process parameters for the PLA and PCL pellets. Then, in the second step, the blends were obtained by means of a Brabender plastograph internal mixing machine (Brabender GmbH & Co. KG, Brabender Plastograph EC, Duisburg, Germany). The working parameters were established at 195 °C, 80 rpm and 10 min. The PLA pellets were dried for 2 h at 80 °C to remove moisture and avoid the lasting hydrolytic degradation effect during the process. Next, the obtained blends were cooled and milled using a blade mill (Retsch SM 100) to crush the material and produce pellets again. As the crushed material was very heterogeneous, ranging from filaments to pellets of several shapes and dimensions, including some larger than what the plasticization chamber of the Sonorus 1G can process, it was sieved to classify the material according to its particle size, and the most suitable size was selected to manufacture the specimens in the USM machine. Four circular sieves (Ø200 mm and 0.8 mm height) were manufactured by additive manufacturing (BCN3D Sigma Release 2017), allowing the classification of the material into four particle ranges (the particle dimensions and the percentage of density flow to each size are shown in Figure 2). As the literature reveals, the more homogeneous and smaller the pellet size is, the better the USM process becomes [33,37]. Thus, the two intermediate particle sizes (referred to as t1 and t2) were chosen to carry out the experimentation. Considering the influence of a rough material shape on USM process parameters [47] and the portion of PLA in the blends, some specimens were manufactured in Step 3 using sieved PLA pellets from Step 2. Finally, these results were taken as a reference to find the process parameters to manufacture specimens of the PLA/PCL blends in Step 4. In summary, seven material types were used to manufacture the specimens: (a) PLA and PCL commercial pellets (referred to as PLA_p_ and PCL_p_), (b) sieved PLA (named PLA_t1_ and PLA_t2_, according to each particle size range) and (c) three PLA/PCL blends. Fifteen specimens were manufactured for each material type.

Table 2 shows the fixed values of the process parameters for all the specimens and those modified during the experimentation (indicated by gray cells). The fixed values were established based on expertise, literature recommendations and some experimental trials. In the compaction, both the force and the strokes values were recommended by S.L. Ultrasion Company (Cerdanyola del Vallès, Barcelona, Spain), whereas the plunger speed was adjusted experimentally. As a result, too high a speed provokes a rebound effect on the pellets, which decreases their compaction. In the melting and injection stages, the force and speed of the plunger were also recommended by S.L. Ultrasion Company (Cerdanyola del Vallès, Barcelona, Spain). The plunger displacement was calculated according to the amount of material needed to fill the cavity, and the injection time was adjusted experimentally by the analysis of the graphical data provided by the machine. In the machine display, the time required to move the plunger from the end of the melting to the start of compaction could be graphically observed; however, due to compaction, there was minimum plunger displacement, and in that sense, 2 s was enough. Regarding the variable process parameters, mainly the amplitude and the ultrasonic time during the plasticizing stage (which takes place between preheating and melting) are included. These are the most influential parameters revealed in the literature to process most materials [29,33] and obtain complete parts, including by studies that processed PLA [35,40], another discussed in the introduction that processed other materials and by the company that developed the machine. In this work, the mold was kept at room temperature.

As mentioned at the beginning of this section, a set of preliminary trials was performed to set the starting process parameters for the PLA and PCL pellets (Step 1 in Figure 2). In the case of PLA_p_, in the preliminary trials, the focus was on the ultrasonic energy provided during the preheating stage of the process to achieve the glass transition status, thus varying the amplitude (28.13, 32.81, 37.50 and 42.75 µm) and the ultrasonic time (1, 2, 3 and 4 s). As Jiang et al. [44] proved, plasticizing an ultrasonic polymer is highly influenced by the initial temperature of the polymer; the closer this temperature is to the glass transition point, the higher the average heating rate becomes. This means that the preheating stage is important when the melting temperature of the polymer is high, such as PLA, notably improving the melting stage of the whole polymer. For PCL_p_, the preliminary trials focused on the melting stage due to it not requiring preheating because of its low melting temperature. A set of amplitudes (18.75, 28.13, 32.81 and 37.50 µm) and times between 1 and 4 s were tested. The results of these preliminary trials were taken next as a reference to establish the final process parameter windows for PLA_t1_, PLA_t2_ and the three PLA/PCL blends.

### 3.3. Characterization Techniques

Analyses of the manufactured specimens included thermal properties, molecular weight variations and chemical degradation behavior by hydrolysis. These results were compared with neat matrices and blends prior to being processed by USM to evaluate the influence of ultrasonic energy. Briefly, characterization was carried out in two different regions of each specimen, as USM may lead to variations depending on the region of the molded material [36,47]. These regions are shown in Figure 1d, leading to the regions A_region_ and B_region_, corresponding to areas close to the injection point and at the end of the specimen, respectively.

The thermal properties of the obtained specimens were assessed by means of differential scanning calorimetry using a DSC Q2000 differential scanning calorimeter (TA instruments, Bellingham, WA, USA) in order to evaluate the influence of ultrasonic energy on the thermal transitions and on the crystallinity degree. Scans were run from 30 to 80 °C and from 30 to 200 °C at a heating rate of 10 °C/min under an inert atmosphere, provided by a constant flow of nitrogen (50 mL/min), for the glass transition temperature (Tg) and the melting temperature (Tm), respectively.

The crystallinity degree was determined both as a function of PLA and PCL, according to Equation (1):(1)$$X_{c}\;\left(\%cristalinity\right)=100\%\;\cdot \;\left[\frac{\Delta H_{m}-\Delta H_{c}}{\Delta H_{m0}}\right]\cdot \frac{1}{W}$$
where Δ*H*_m_ is the enthalpy of melting (corresponding to the fusion process), Δ*H*_c_ is the crystallization enthalpy, Δ*H*_m0_ is the enthalpy value of a pure crystalline material and W is the weight fraction of PLA or PCL in the blend. The reference value for the Δ*H*_m0_ of PLA was 93 J/g [48] and 139 J/g for PLC [48,49].

The gel permeation chromatography (GPC) technique was used to investigate the molecular weight by means of a Waters 410 refractometer index detector and a Waters 600E pump connected to a Styragel HR column. The samples were prepared by dissolving between 20 and 25 mg of specimen in 5 mL of filtered tetrahydrofuran (THF) solvent. Molecular weight was determined both in neat matrices and blends and the obtained specimens by means of USM, as well as in the two abovementioned regions (A_region_ and B_region_) of the final specimen for comparison purposes. Thus, the influence of ultrasonic energy and its variability on the part could be observed.

Finally, an in vitro accelerated degradation test was performed to analyze the chemical degradation rate of the PLA/PCL blends processed by USM, mimicking conditions suitable for medical applications. The degradation test was evaluated by the weight loss experienced by the specimens into two different media: (1) acidic, using a hydrochloric acid solution at pH 3, similar to the physiological stomach conditions; and (2) basic, by phosphate-buffered solution with pH 7.4, simulating the conditions of saliva or blood, over a period of 180 days (6 weeks) and taking measurements on Weeks 2, 4 and 6. For this study, three replicates of each region of the manufactured specimens were analyzed for each solution (10–29 mg). The weight was measured using Mettler Toledo XS3DU weighing scales (capability of 0.8 g/3.1 g and repeatability of 1 μg/10 μg), and the weight loss was obtained using Equation (2):(2)$$Weight\;loss\;\%=\frac{W_{o}-W_{d}}{W_{o}}\cdot 100$$
where *W*_o_ is the mass sample before, and *W*_d_ is the mass after being added into the solution [50,51].

## 4. Results and Discussion

### 4.1. Process Parameter Windows

Table 3 shows the process parameter windows adopted in USM to each specimen type, starting from the results of the preliminary trials presented in Section 3 for the PLA and PCL pellets. In the case of PLA_p_, the successful combinations were amplitudes of 32.81–42.75 µm vibrating for 2–3 s, but the minimum values of time and amplitude that provided completed parts were selected. At higher time values, material degradation was observed, while at softer conditions, the mold was not successfully filled (Figure 3a). These results align with the literature values [34,35], in which different machine configurations were used, and the acoustic unit was almost the same. For PCL_p_, a minimum of 32.81 µm and 2.5 s were required to fill the mold. For the sieved PLA (PLA_t1_ and PLA_t2_), the ultrasonic melting time (taken from PLA_p_) was reduced, as the material burned and degraded. This effect might be attributed to the irregularities in size and shape compared to commercial pellets so that they count on a higher specific surface area, which may promote the heat transfer and friction heat effect [52]. The time was set between 0.8 and 1 s, where full mold filling was observed. The minimum time was also selected (0.8 s). The blends were processed according to the process parameters of PLA_t_, reducing the preheating time due to the presence of PCL (Figure 3b). Figure 3c shows the specimens of the PLA/PCL blends.

Although knowing the polymer temperature would be useful to study the influence of the process parameters over the raw material and the blends on the plasticizing behavior, the high speed of the process complicated its acquisition in this research, as the sensor presented a low measurement speed. In substitution, the ultrasonic energy was determined.

Figure 4 plots the ultrasonic energy provided by the generator for each material to manufacture the specimens, including both energies provided during preheating and melting. This energy is related to the thrust that the acoustic unit receives by means of the sonotrode from the material that is being pushed by the plunger to fill the mold cavity, so it can be assumed that the energy provided equals that received by the material. Thus, when the material flows better, entering the mold more easily, the sonotrode receives less effort and therefore provides less energy. On the contrary, the energy will be greater due to the higher the thrust it receives from the bottom. This thrust depends on the kinematics of the plunger caused by the process parameters of force and speed, which remained the same for all the material types, and the material flow capability due to the action of the ultrasounds. As expected, at higher amplitudes and/or ultrasonic times, higher energy was measured.

The influence of the pellet size and shape is observed by comparing the energy of the processing specimens of the sieved PLA (PLA_t1_ and PLA_t2_), where the process parameters were exactly the same. At smaller pellet sizes (PLA_t1_), less melting energy is required to obtain complete parts (around 13% lower than PLA_t2_), bringing to light the influence of interfacial friction heating in the initial seconds of the ultrasonic process [45], which increases the polymer temperature in the preheating and improves polymer fluidity during the melting stage. This also explains the reduction in the ultrasonic time between PLA_p_ and PLA_t1&t2_ from 1.2 s to 0.8 s during melting, as the commercial pellets are bigger and more regular. This phenomenon contributed to the results provided by [47], who proved that the pellet shape is statistically significant on the manufactured specimen weight, providing heavier parts when more irregular pellets were used, probably caused by the improvement in the fluidity due to increment in temperature as a consequence of the interfacial friction heating.

Comparing the blends with each other, no significant differences were observed, and the decrease in the ultrasonic preheating time, compared to the sieved PLA, also brought the preheating energy down. However, the total energy for processing the blends was higher. This effect may come from the achieved temperature by the polymer during preheating, which might be lower; thus, the viscosity might be higher, requiring more energy.

### 4.2. Thermal Properties

The main thermal properties and characteristics of each neat polymer and blend are reported in Table 4, including the melting temperature (T_m_), the crystallization temperature (T_c_), the glass transition temperature (T_g_) and the corresponding enthalpies (melting and crystallization enthalpy, ${\Delta \hat{H}}_{m}$ and ${\Delta \hat{H}}_{c}$, respectively) to calculate the crystallinity degree of the polymers, both pellets and specimens in the two selected regions.

It is worth noting that both the T_m_ and T_g_ of all the studied materials were not affected by ultrasonic energy, exhibiting maximum differences around 2 °C before and after being processed, although differences were found in the crystallinity degree. Indeed, this is in agreement with previously published results [34].

USM decreased the PLA crystallinity (up to 25%) and increased it when PCL was processed (around 10%). For the sieved PLA (PLA_t1_ and PLA_t2_), the thermal behavior compared to the PLA commercial pellets remained constant, although X_c_ fell by more than 65% after being processed (the appreciation is shown graphically in Figure 5a). Such a decrease in the PLA crystallinity is attributed to its low crystallization rate, which is slower than the cooling rate applied in the most conventional processing techniques, leaving a low crystalline material that is often transparent [53]. In this case, using USM, this effect was increased due to the small size of the samples, which can perform the cooling process faster than bigger parts, often produced by conventional processes. This result can be suggested after the observation of Figure 4a, where the PLA samples were characterized by DSC before and after USM processing. It is easy to observe that the raw material did not experience crystallization during the DSC test (no exothermal signal around 100 °C), indicating a crystalline material at the highest capacity. However, after USM processing, the material crystallized during DSC, as PLA could not crystallize at maximum capacity during the cooling time. Moreover, this lack of PLA crystallization caused by its low crystallization rate and the high speed of the USM process and cooling leads to a material with higher content of glassy phase (noncrystalline), allowing the observation of a T_g_ close to 60 °C (Figure 4a). It is also possible to observe some physical aging of PLA, linked to the small endothermic peak together with the T_g_ [54]. Moreover, the phenomenon of fractional crystallization can also be seen in the DSC diagrams. This crystallization can be directly attributed to the high cooling rate during USM processing, so the material may not have enough time to crystallize. For this reason, at low heating rates, the samples crystallized, and this peak appears on the graph (Figure 6a).

In the case of PCL, crystallinity increased (up to 10%) after being processed due to its inherent high crystallization rate, with USM processing faster than injection molding (Figure 5b). Therefore, nontransparent parts were obtained. As mentioned above, small specimens have the capacity to cool quickly. Taking into account that the T_c_ of pure PCL is close to room temperature (22–24 °C) as previously reported [55], the samples produced by USM reached room temperature very fast, reaching optimal crystallization conditions faster and leaving materials with a higher crystalline phase.

The crystallinity and thermal behavior along the specimen (meaning the comparison between A_region_ and B_region_) remained almost constant in both polymer matrices (Figure 6a), although the PLA crystallinity decreased slightly at the end of the specimen (less than 11%), due to the faster cooling of the material in this region, and on the contrary, it increases slightly for PCL (less than 5%). Comparing the diagrams of the molded specimens (Figure 6a), no significant differences were detected between the two studied regions (A_region_ and B_region_), which indicates that the thermal properties were constant throughout the specimen.

Regarding the PLA/PCL blends, the trends of the crystallinity of each component of the blends were quite similar to those obtained in the neat matrices, decreasing for PLA and increasing for PCL (Figure 5). However, the decrease in the PLA crystallinity with respect to the virgin material in the presence of low contents of PCL (up to 30%) could be derived from the interruption of the crystallization of PLA from the PCL molecule during cold crystallization [24]. No clear trend was observed depending on the composition of the blend, obtaining the same result as that previously reported [8] due to the low miscibility between both components in the blend.

Figure 6b reveals the immiscibility between both polymers, as no variations in T_m_ were obtained compared to those of individual materials. Again, the crystallization of PLA was observed during DSC, related to the lower capacity of this polymer to crystallize during the cooling stage of USM processing.

### 4.3. Molecular Weight

Figure 7 shows the obtained molecular weights from the GPC technique, including the number average molecular weight ($\bar{M_{n}}$) and the polydispersity index (PI), which are used to measure the amplitude of the molecular weight distribution.

Comparing the PCL and PLA material before and after being processed, USM increased both the $\bar{M_{n}}$ and the PI. Thus, the ultrasound vibration energy induced chemical reactions, leading to crosslinking between the molecular chains of the material [39] or molecular segregation caused by the integration of lower-molecular-weight chains that melted first and incorporated into the small regions acting as joints between pellets [34]. When single polymers were processed, the $\bar{M_{n}}$ decreased from the beginning to the end of the specimen (13% in PLA and 3% in PCL), whereas the PI increased (28% in PLA and 12% in PCL) (Figure 7), which agrees with the trend found in the literature [47]. This means that molecular chain scissions occurred during the cavity filling, probably because of the movement produced by the ultrasonic vibration [39]. However, when the blends are analyzed, the trend is the other way around, so that both $\bar{M_{n}}$ and PI increased from the beginning to the end (Figure 7), this increment being influenced by the PCL portion into the blend. This phenomenon could presumably be explained by the excess of energy that PCL receives due to the absence of a preheating stage, as during this process, the polymer degrades, disentangling and breaking molecular chains, producing the homolytic cleavage of C-C and/or C-H and delivering radicals that react with each other, affecting the molecular weight.

### 4.4. Degradation Test

As stated above, PLA and PLC are strong alternatives to fossil-based materials, particularly for biomedical devices that might be processed by means of USM. Understanding their degradation rate in similar conditions to those of the human body might provide useful information for determining the final application of the obtained products. Thus, Figure 8 shows the weight loss that the samples experienced over time (2, 4 and 6 weeks) in the two different media: basic (pH 7.4) and acidic (pH 3). As expected, the degradation ratios of both materials were different, with the PCL material suffering the greatest weight loss in both media (Figure 8a,b). PCL degraded faster in basic conditions than in acidic, although in both cases, the degradability level was slightly lower than others reported in the literature [56]. This might be caused by the moderate level of solutions, as according to [56], the stronger the conditions are (both acidic and basic), the higher the degradability ratio becomes. The weight loss for PLA was more pronounced under acidic conditions, although two clear trends were detected (Figure 8b). During the first two weeks, a more or less linear degradation profile was observed, which continued with a lower slope during the weeks after, in the same way as previously reported [51]. On the other hand, the PLA/PCL blends exhibited a weight loss according to the PLA/PCL ratio and the media (Figure 8c,d). Hence, the higher the PCL portion was in a basic medium, the higher the degradation became, this effect being less pronounced in an acidic medium. When the PLA/PCL blends were in basic conditions (Figure 8d), the weight loss was dominated by the PCL trend, although it occurred after the second week. On the contrary, in acid conditions, the PLA portion counteracted a trend of weight loss (Figure 8c). Therefore, this confirms that blends allow the production of a new class of materials “on demand”, with fit-for-purpose and tailored characteristics to their final application.

## 5. Conclusions

In this work, USM technology was used to analyze the feasibility of manufacturing PLA/PCL blends, providing the process parameter windows to obtain the manufactured parts and characterize the effect of the ultrasound on these parts from thermal, chemical and degradation points of view.

The process parameter windows were gradually obtained, considering two relevant issues. On the one hand, considering conditions more energy efficient for neat matrices (PLA and PCL) and, on the other hand, the variability of both pellet shape and size coming from the blended material. As a main result, it is worth noting that less ultrasonic energy was required to melt the irregular pellets due to the heating increment caused by the interfacial friction, which reduced the ultrasonic exposure time of the polymer from 1.2 s to 0.8 s during the melting stage, saving around 15% energy (in comparison to the industrial pellets with a rounded shape). Moreover, the pellet size in the range of 1.5–3.8 mm was the most efficient, followed by those within the range of 3.8–4.3 mm.

From a thermal point of view, USM technology kept both the T_m_ and T_g_ of the polymers constant. However, effects on crystallinity were found, experiencing a notable decrease in PLA (up to 25%) and a slight increase in the case of PCL (around 10%), these trends being slightly lower for the blends. Of course, it depends on the material’s crystallinity ratio, the quick speed of the process and the small geometry dimensions. Certainly, USM technology is focused on small parts; however, by properly modifying the melting speed and increasing the cooling time, these values could be modified and their influence on the mechanical properties assessed. This will be performed in future work.

As expected, ultrasonic energy influenced the $\bar{M_{n}}$ and the PI of the processed polymers and through the processed specimen (from the beginning to the end). The USM process increased the $\bar{M_{n}}\;$ by 15% and the PI by 30% for the PCL and PLA material before and after being processed, which could suggest a partial polymer degradation. These results contrast with the literature results, where the $\bar{M_{n}}$ trend is reduced after being processed [34,35,47], which may suggest that the ultrasonic energy provided to the polymer could be optimized. Moreover, the molecular weight changed from the beginning to the end of the specimen, decreasing in the neat polymer and increasing in the blends, whereas the PI increased in both cases. This could be attributed to chain scissions due to ultrasonic vibration from one side and by the reaction of free radicals of PCL instigated by the excess of energy provided during the processing of blends from the other side. In the future, the process parameters should be adjusted to provide less ultrasonic energy and avoid these structural changes to the processed polymers.

Finally, the weight degradation ratio is in accordance with the literature trends, with the PCL ratio being higher than the PLA in both media and faster in basic than in acid conditions.

Polymer blends captured the attention of this research due to the easy and reasonable way to expand polymer properties. Focusing on the development of high-performance polymers, these types of blends should be obtained with a small batch size, which agrees with the advantages of USM technology. Once the feasibility of this successful match has been proved, further studies should cover the optimization of process parameters, testing different blends and extending the material characterization.

## Figures and Tables

**Figure 1 polymers-13-02412-f001:**
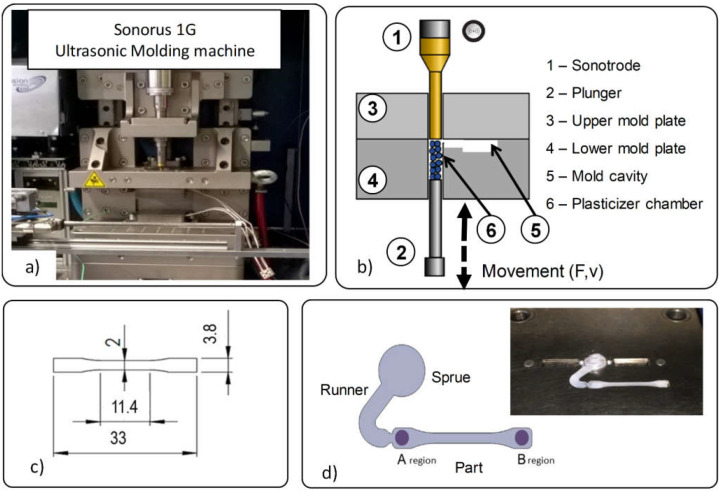
Ultrasonic molding technology: (**a**) Sonorus 1G machine, (**b**) main elements, (**c**) specimen and (**d**) specimen regions: start (A) and end (B).

**Figure 2 polymers-13-02412-f002:**
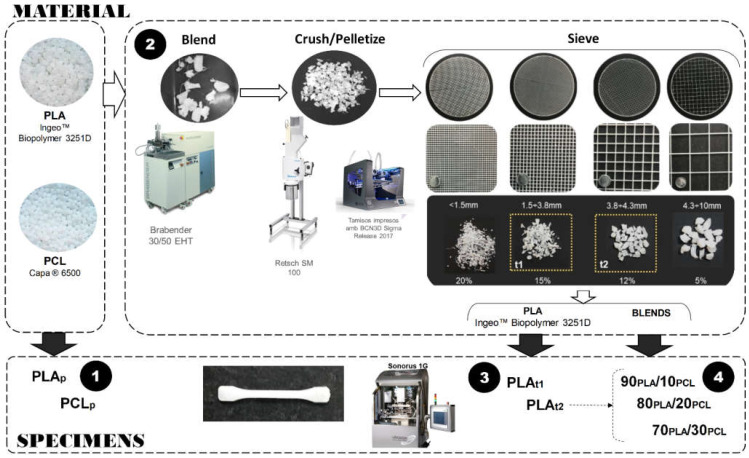
Manufacturing process chain.

**Figure 3 polymers-13-02412-f003:**
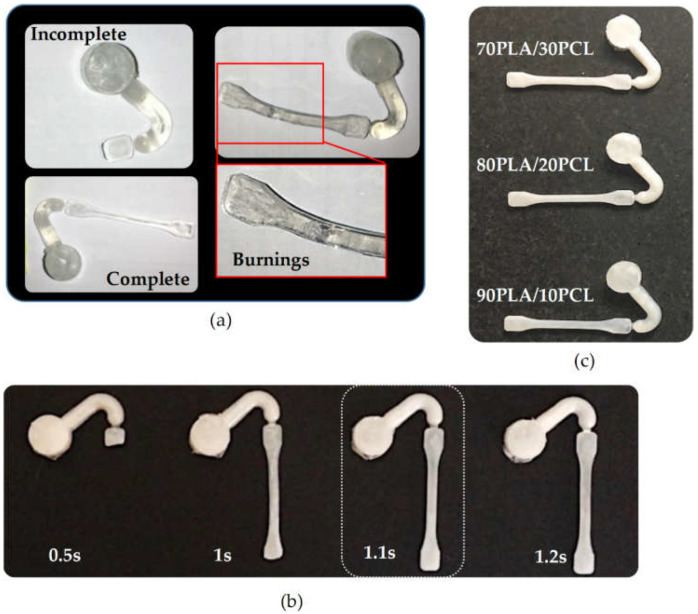
Manufactured specimens: (**a**) PLA preliminary trials, (**b**) influence of preheating time on 90PLA/10PCL specimens and (**c**) specimens of PLA/PCL blends.

**Figure 4 polymers-13-02412-f004:**
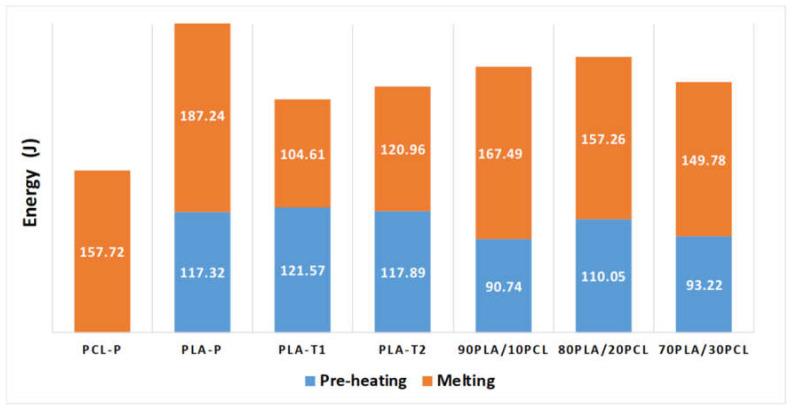
Ultrasonic energy for manufacturing the specimens.

**Figure 5 polymers-13-02412-f005:**
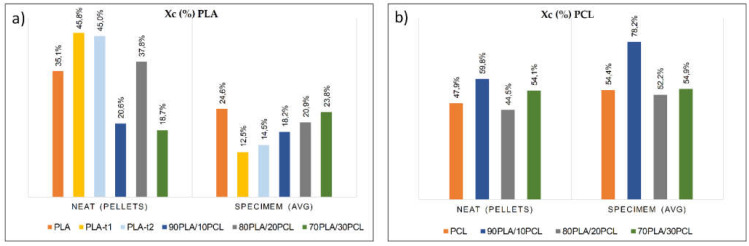
Crystallinity changes: (**a**) X_cPLA_ and (**b**) X_cPCL_.

**Figure 6 polymers-13-02412-f006:**
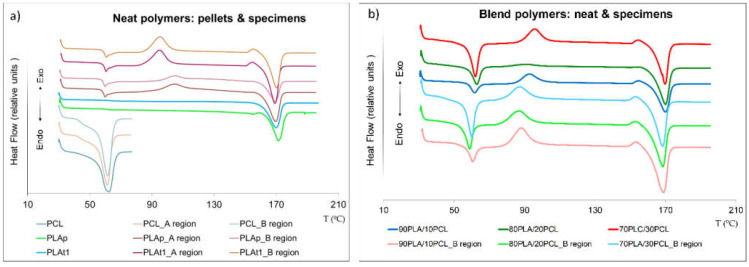
DSC plots: (**a**) neat polymer and (**b**) blend polymer, before (pellets) and after (specimens) being processed by USM.

**Figure 7 polymers-13-02412-f007:**
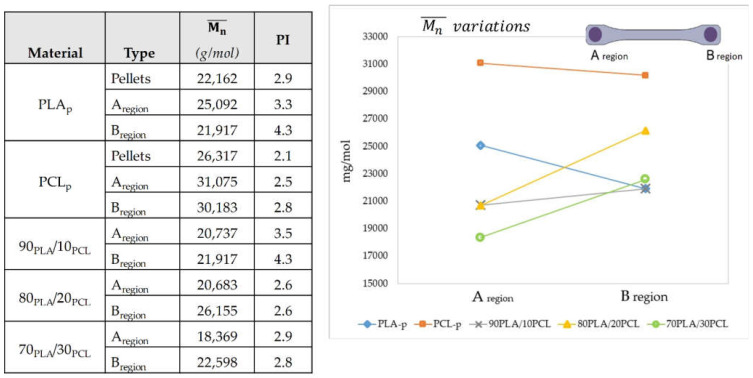
$\bar{M_{n}}$ and PI variations of the obtained pellets and specimens.

**Figure 8 polymers-13-02412-f008:**
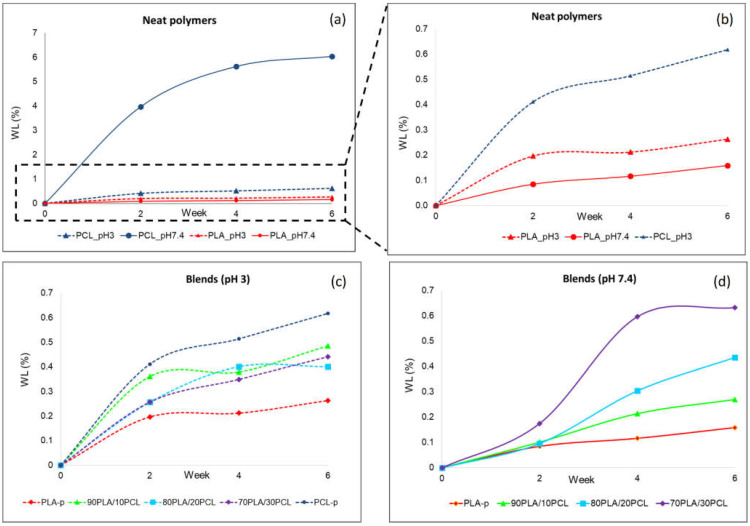
Degradation in terms of weight loss: (**a**) neat polymers, (**b**) extension of neat polymers, (**c**) blend degradation at pH 3, and (**d**) blend degradation at pH 7.4.

**Table 1 polymers-13-02412-t001:** Process parameters in ultrasonic molding technology.

Phase	Vibrational Amplitude (A, µm)	Plunger Force (F, N)	Plunger Speed (v, mm/s)	Time (t, s)	Plunger Displacement (D, mm)	Strokes (St)
Compaction		✓	✓			✓
Preheating	✓	✓		✓		
Melting	✓	✓	✓	✓	✓	
Injection/packing		✓	✓	✓		
Cooling				✓		

**Table 2 polymers-13-02412-t002:** Process parameters in the ultrasonic molding tests.

Phase	Vibrational Amplitude (A, µm)	Plunger Force (F, N)	Plunger Speed (v, mm/s)	Time (t, s)	Plunger Displacement (D, mm)	Strokes (St)
Compaction		4000	10			5
Preheating	--	--		--		
Melting	--	625	1	--	2.1	
Injection		3000	1	2		
Cooling				5		

**Table 3 polymers-13-02412-t003:** Process parameters windows: preheating and melting.

	Variable Process Parameters							
	Parameters	PLA_p_	PCL_p_	PLA_t1_	PLA_t2_	90PLA/10PCL	80PLA/20PCL	70PLA/30PCL
Preheating	Vibrational amplitude (A, µm)	37.50	0	37.50	37.50	37.50	37.50	37.50
	Plunger force (F, N)	1000	0	1000	1000	1000	1000	1000
	Time (t, s)	2	0	2	2	1.1	1.4	1.3
Melting	Vibrational amplitude (A, µm)	46.88	32.81	46.88	46.88	46.88	46.88	46.88
	Time (t, s)	1.2	2.5	0.8	0.8	0.8	0.8	0.8

**Table 4 polymers-13-02412-t004:** DSC output data.

Material	Material Type	T_m PCL_ (°C)	T_g PLA_ (°C)	T_cc1 PLA_ (°C)	T_cc2 PLA_ (°C)	T_m PLA_ (°C)	H_m PCL_ (kJ/mol)	H_cc1 PLA_ (kJ/mol)	H_cc2 PLA_ (kJ/mol)	H_cc PLA_ (kJ/mol)	H_m PLA_ (kJ/mol)	X_cPLA_ [%]	X_cPCL_ [%]
PLA_p_	Pellets		61.4	159.3		171.0		0.7		0.7	33.6	35.1	
	A_region_		59.0	104.5		169.6		14.6		14.6	38.8	25.8	
	B_region_		58.8	104.3		169.4		9.0		9.0	30.9	23.4	
PCL_p_	Pellets	62.2					66.5						47.9
	A_region_	61.1					73.6						52.9
	B_region_	61.5					77.7						55.9
PLA_t1_	Pellets		61.9			169.8				0.0	42.9	45.8	
	A_region_		59.8	95.0	154.4	169.0		26.9	3.0	29.9	42.4	13.3	
	B_region_		59.7	95.1	155.1	170.2		27.8	2.5	30.3	41.3	11.8	
PLA_t2_	Pellets		60.6			170.0				0.0	42.1	45.0	
	A_region_		60.1	93.8	154.4	169.4		25.1	2.6	27.6	42.8	16.2	
	B_region_		60.3	94.6	155.0	169.6		26.0	3.0	29.0	41.1	12.9	
90_PLA_/10_PCL_	Pellets	62.9	61	90.7	154.5	169.9	12.4	3.3	0.1	3.4	31.7	37.8	44.5
	A_region_	60.0	58.3	86.7	153.9	168.9	14.0	14.4	1.5	15.9	34.7	25.0	50.3
	B_region_	59.9	57.6	86.7	152.9	168.6	17.0	18.6	1.5	20.1	33.9	18.3	61.2
80_PLA_/20_PCL_	Pellets	62.1	60.7	95.8	154.5	169.8	22.6	16.4	1.8	18.2	30.5	18.7	54.1
	A_region_	60.3	58.6	85.9	153.8	168.6	23.7	13.8	1.7	15.5	30.7	23.1	56.8
	B_region_	60.1	58.5	87.0	153.5	168.3	23.1	14.9	2.0	16.9	30.8	21.1	55.5
70_PLA_/30_PCL_	Pellets	62.9	61	90.7	154.5	169.9	12.4	3.3	0.1	3.4	31.7	37.8	44.5
	A_region_	60.0	58.3	86.7	153.9	168.9	14.0	14.4	1.5	15.9	34.7	25.0	50.3
	B_region_	59.9	57.6	86.7	152.9	168.6	17.0	18.6	1.5	20.1	33.9	18.3	61.2

## Data Availability

Data is available upon request to the corresponding author.

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
