# Peer review of "Manufacturing PLA/PCL Blends by Ultrasonic Molding Technology"

_polymers, 2021, doi:10.3390/polym13152412_

Round 1

Reviewer 1 Report

The manuscript seems very interesting, both the abstract and the introduction, methodology, discussion of the results and conclusion fit the Polymers target. However, I would like to know why the characterization of the degradation in medical application has been chosen and for instance the improve of compatibility or affected functional groups due to degradation produced by USM by means of FTIR has not been analyzed. On the other hand, I miss the comparison of the USM technique with the most used molding techniques (injection), in order to compare parameters like:  energy used to manufacturing the specimens, time, others.

Author Response

Please see the attachment file. The authors wish to thank the Reviewer the provided feedback

Reviewer 2 Report

  1. The manuscript is well organized, and the subject is well presented. The methods used are sound and the presentation and discussion of results is logical.
    The manuscript requires some major revisions to bring it to a level worthy of publication. My recommendations are detailed below:
  2. The current study investigates the use of ultrasonic molding and its capability for producing PLA/PCL blends. For this, the authors study the effect of pellets shape and dimensions in addition to the manufacturing process parameters. The authors analysed the thermal characteristics and found that using the ultrasonic process does not affect them. In addition, the authors found that crystallinity rate is somewhat affected and increase the molecular weight.
  3. The abstract is acceptable but can be improved, please improve it by answer all the parts in the following question: Please consider reviewing the abstract and highlight the novelty, major findings and conclusions.
  4. Please consider removing or reducing the length of 2. UltraSonic Molding technology. This section is more suitable in a thesis chapter or a report but not in a scientific paper, Information on what is ultrasonic molding process and how it works and more details can be found about it from books and other types of reports.
  5. Before the end of the introduction the authors should attempt to answer the following question: What is the research gap did you find from the previous researchers in your field? Mention it properly. It will improve the strength of the article.
  6. At the end of the introduction the authors need to briefly summarise again what was done in this work.
  7. Line 175-180 please combine all in one paragraph, please avoid writing small paragraphs of 1-3 lines only.
  8. In Table 2, why the authors choose those specific parameters? Are they based on past literature or industry recommendation or from the authors own knowledge? In line 214 you said its from literature and cited 3 references, but does these three references studies all the seven materials which are investigating in this work?
  9. The authors should make sure that all the table formats are the same (For example Table 2 and Table 3 have different font style and different design).
  10. The authors should add a list of nomenclature at the end of the manuscript for all the Greek letters, symbols and abbreviations used in this study.
  11. What is the number of samples tested for each of the materials analysed in this study?
  12. It is strongly recommended that the authors include a table in the manuscript which contains some of the mechanical and thermal properties of each of the seven types of materials used in this study from the open literature (add it in the materials and methods section in the first section).
  13. It is advised to move Table 3 to the materials and method section instead of the results and discussion section unless the authors have a good reason for that.
  14. Line 311-312 can the authors add some images of the manufactured specimens?
  15. Lines 326-328 can the authors explain why the melting and glass temperatures did not get affected by the ultrasonic energy, what is the fundamental theory behind this, and how about past studies in the open literature, did they find similar results or different from yours? In either way, please discuss and support with references.
  16. Table 4 would be much better presented in bar chart graphs (suggestion).
  17. Line 359-360 “, although a slight increase at the end in PLA it increases lightly at the end due..” please check this sentence it does not read well, also what is the difference between lightly and slightly, it is a bit confusing, please make sure to use scientific words to describe a quantity or some results, for example consider using %, say increase by 2% only to show that the increase is slight or light! Or something similar to that. Please check this issue elsewhere in the manuscript.
  18. Figure 4 where the y axis numbers? Or they are not supposed to have any? I see heat flow but there is not numbers to indicate the values of the heat flow for each of the material at different temperatures?
  19. Figure 5 the right side one the image on the top right is not clear, please enlarge it what is this?
  20.  The authors are encouraged to include more detailed discussion and critically discuss the observations from this investigation with existing literature.

Author Response

Please see the attachment file.  The authors wish to thank the Reviewer for the  provided feedback. 

Round 2

Reviewer 2 Report

All qustions answered, however please improve the quality of all figures 

Author Response

Thanks. 
